# Harnessing mechanical instabilities at the nanoscale to achieve ultra-low stiffness metals

Samuel Temple Reeve [1], Alexis Belessiotis-Richards[2] & Alejandro Strachan[1]

Alloy and microstructure optimization have led to impressive improvements in the strength of engineering metals, while the range of Young's moduli achievable has remained essentially unchanged. This is because stiffness is insensitive to microstructure and bounded by individual components in composites. Here we design ultra-low stiffness in fully dense, nanostructured metals via the stabilization of a mechanically unstable, negative stiffness state of a martensitic alloy by its coherent integration with a compatible, stable second component. Explicit large-scale molecular dynamics simulations of the metamaterials with state of the art potentials confirm the expected ultra-low stiffness while maintaining full strength. We find moduli as low as 2 GPa, a value typical of soft materials and over one order of magnitude lower than either constituent, defying long-standing composite bounds. Such properties are attractive for flexible electronics and implantable devices. Our concept is generally applicable and could significantly enhance materials science design space.

[1] School of Materials Engineering and Birck Nanotechnology Center, Purdue University, West Lafayette, IN 47906, USA. [2] Department of Materials, Imperial College London, Exhibition Road, London SW7 2AZ, UK. Samuel Temple Reeve and Alexis Belessiotis-Richards contributed equally to this work. Correspondence and requests for materials should be addressed to A.S. (email: strachan@purdue.edu)

Engineered composites combine distinct materials to achieve properties or processability not available in the individual components and find widespread use, from advanced airframes[1] to biomedical implants[2]. Nature also uses composites to achieve outstanding mechanical performance by arranging relatively humble materials in optimized microstructures[3]; classic examples include nacre[4], bone[5], and spider silk[6]. Unfortunately, the design of composites is not without limitations. With few notable exceptions, their properties fall between those of its constituents and various forms of the rule of mixtures play a central role in composite design. The weighted average rule applies rigorously for properties like density where the value for the composite can be obtained as a sum over phases: $\sum x_i \rho_i$ where $x_i$ is the volume fraction of phase $i$ and $\rho_i$ its density. Properties like stiffness depend on the arrangement of phases and elegant solutions exist for simple geometries[7]. Such properties are bounded by extrema obtained by the addition of phases in series or in parallel. This is true even as composite, foam, and lattice material design has successfully filled previously inaccessible materials property space[8], for example, ultra-low stiffness metallic micro-lattices[9, 10] and structures with negative Poisson's ratio[11, 12].

There is significant interest in breaking rule of mixtures constraints to reach otherwise unachievable properties. Breakthroughs in materials science and engineering have enabled overcoming such constraints for extrinsic properties, those that depend strongly on microstructure. Examples include biomimetic composites with high toughness[13, 14] and composites designed for low thermal conductivity[15]. Intrinsic properties have proven more challenging. We mentioned that the weighted average rule applies exactly to density and has, to date, provided inexorable bounds on stiffness. Even the recent, remarkable observation of stiffening of a solid via nanoscale liquid inclusions can be explained by classic composite theory after the incorporation of interfacial tension[16].

In this paper, we use theory to design an ultra-low stiffness metamaterial, a composite that defies the constraints of the rule of mixtures for a quasi-static property weakly sensitive to microstructure: stiffness. We note that stiffness depends on microstructure (via texture), but to a much lesser degree than strength and toughness, and can to date only be engineered to a large extent via open structures such as foams and lattices, which incorporate air as a second phase. Molecular dynamics (MD) simulations that explicitly model the mechanical metamaterial[17] show a stiffness over one order of magnitude smaller than that of its softest component. In absolute terms, we show a fully dense metal with a stiffness of as low as 2 GPa, a value typical of polymers. This unprecedented result is possible because one of the components in the metamaterial is stabilized in a thermodynamically forbidden state of negative stiffness by interfacial stresses originating from its coherent integration with a thermodynamically stable material. This idea of harnessing unstable states has been demonstrated experimentally using ferroelectrics in nanoelectronic applications[18, 19] and to achieve both ultra-high damping[20, 21] and ultra-high stiffness[22] composites. While these studies have produced impressive properties, most are dynamic and the displayed properties are transient; ref. [18] is an exception for electronic properties. In contrast to prior efforts in mechanical properties, the metamaterials described in this paper stabilize a negative phase, resulting in ultra-low stiffness under quasi-static conditions.

## Results

**Ultra-low stiffness metamaterial design.** Materials experience thermo-mechanical instabilities during phase transformations:

**Fig. 1** Free energy landscape engineering in NiAl. Free energy as a function of in-plane lattice parameter from MD simulations of bulk $Ni_{63}Al_{37}$ and NiAl, interpolated with rule of mixtures. The combination of 65 at% $Ni_{63}Al_{37}$ (thick red) and 35 at% NiAl (thick blue) is highlighted in black to show the possibility of ultra-low stiffness. The martensite and austenite are labeled for $Ni_{63}Al_{37}$. The inset shows a schematic of the epitaxial interface between NiAl and $Ni_{63}Al_{37}$ where Al atoms are light blue, Ni in NiAl is dark blue, and Ni in $Ni_{63}Al_{37}$ is red. Note that this atomic coloring is unique to this figure

classic examples are phase separation via spinodal decomposition[23] and martensitic transformations that govern shape memory and superelasticity[24, 25], as well as strengthening in some advanced alloys[26, 27]. We harness martensitic transformations, solid–solid phase transformations where a material switches between a high-temperature and high-symmetry (often cubic) phase called austenite to a low-temperature, lower-symmetry phase martensite. The transformation does not involve atomic diffusion and is often accompanied by significant changes in lattice parameters.

The design of ultra-low stiffness metamaterials starts with the free energy landscape that governs a martensitic transformation, that is, free energy as a function of a transformation progress variable. We find it convenient to use a lattice parameter as the progress variable. The red curve in Fig. 1 shows the energy landscape underlying the transformation from a cubic to a monoclinic phase through the application of biaxial strain to a model disordered metallic alloy that approximately describes a $Ni_{63}Al_{37}$ alloy. The details of the model will be described below; we now focus on general aspects of the metamaterial design. The energy landscape exhibits two local minima. One minimum represents the ground state structure, which for this system at room temperature is the martensite; a large energy barrier separates this ground state from a metastable state (austenite in our case). Materials scientists make use of both ground states and metastable phases, the latter being feasible as long as high-energy barriers ensure their long-term viability. In contrast, we are not interested in using either local minima, but instead the states around the local maximum. The local curvature of the free energy vs. lattice parameter represents an elastic constant (the stiffness of the material in response to deformation represented by the progress variable) and the states around the maximum exhibit negative stiffness. That is, under these conditions the material is at or near a mechanical tipping point. Thermodynamics tells us that such a state is unstable[28] and our goal is to harness it to achieve an ultra-low stiffness metamaterial by stabilizing it, barely, at the atomic scale using interfacial stresses originating from its epitaxial integration with a second phase of positive stiffness, represented by the blue line in Fig. 1. An epitaxial or coherent interface involves no defects and atomic planes in one phase continue into the other; both materials are therefore forced into sharing the same lattice parameters along the interfacial

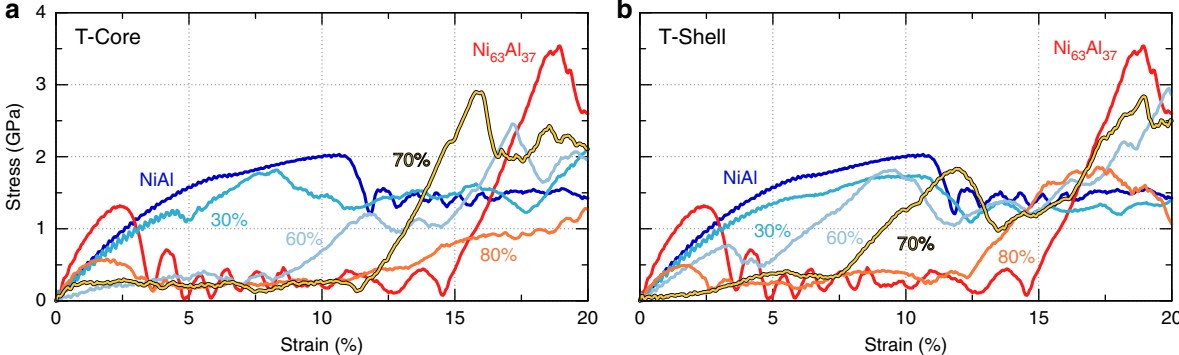

**Fig. 2** Stress-strain response for metamaterial nanowires. Representative (**a**) T-Core and (**b**) T-Shell nanowires, where homogeneous systems are labeled as NiAl and $Ni_{63}Al_{37}$ and metamaterials are identified by composite fraction of the $Ni_{63}Al_{37}$ phase. The lowest stiffness cases occur for 60 at% T-Core in **a** and 70 at% T-Shell in **b**

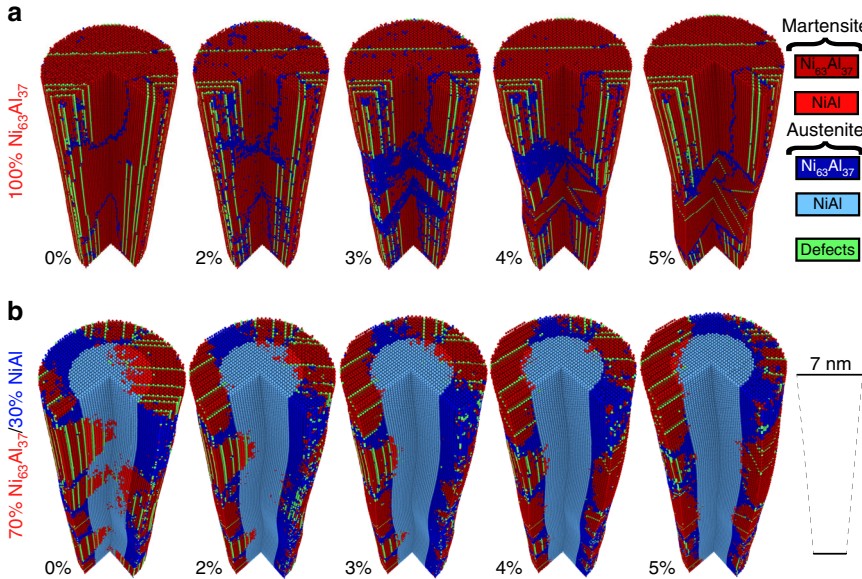

**Fig. 3** Atomic structures contrasting homogeneous and metamaterial nanowires. **a** The homogeneous $Ni_{63}Al_{37}$ nanowire accommodates uniaxial strain through nucleation of a new domain orientation. **b** 70 at% T-Shell metamaterial nanowire changes the local lattice parameter, fraction of martensite and austenite in each region, and relative fraction of multiple martensitic domains in response to strain. Atoms colored by phase through CNA; surface atoms were removed for clarity, as were a front quarter-section of atoms

plane. Thus, to describe epitaxial metamaterials, we modify the rule of mixtures and add the free energies of the two phases at each lattice parameter weighted by volume fractions. The thin lines in Fig. 1 show the results of this new epitaxial rule of mixtures for various phase fractions. Some of the resulting energy landscapes exhibit the desired low curvature (one is highlighted in black), indicating the possibility of ultra-low stiffness.

**Molecular dynamics description of shape memory.** We now demonstrate the feasibility of the approach using large-scale MD simulations with an atomistic model developed to describe $Ni_xAl_{1-x}$ alloys, which exhibit martensitic transformation and shape memory for x between 60 and 65 at%[29]. The interatomic potential is a widely used embedded atom model (EAM) developed by Farkas et al.[30], which, as in the real case, exhibits martensitic transformation from a B2 (CsCl) to a monoclinic phase in Ni-rich $Ni_xAl_{1-x}$. The martensitic transformation of our model system involves the expansion of two of the cubic lattice parameters, taken as the order parameter, and the contraction of the third together with a change of the monoclinic

angle. To stabilize the $Ni_{63}Al_{37}$ alloy in its unstable state (Fig. 1), we need a material with equilibrium lattice parameter (energy minima) around 2.9 Å. It turns out that the equiatomic NiAl-ordered alloy (stable only in the B2 phase) fits the bill, see blue curve in Fig. 1. We therefore explored the possibility of ultra-low stiffness with MD simulations of heteroepitaxial metamaterials consisting of disordered $Ni_{63}Al_{37}$ and B2 NiAl organized in core/shell nanowires and nanolaminate configurations.

Figure 2 shows stress-strain curves of a representative series of core/shell nanowires with various fractions of each phase. The simulated wires are initially 14.5 nm in diameter and 58 nm in length, and uniaxially deformed at a strain rate of $5 \times 10^8\ s^{-1}$ at 300 K; see "Methods" section for additional details. The two homogeneous wires, NiAl and $Ni_{63}Al_{37}$, show the baseline properties of the model material and are discussed first. The pure NiAl wire exhibits the expected behavior for a defect-free nanowire: non-linear elasticity followed by transformation to a strain-stabilized martensitic variant (due to the high cost of nucleating dislocations) with significant stacking faults, see Supplementary Fig. 1. The $Ni_{63}Al_{37}$ wire begins in the martensitic

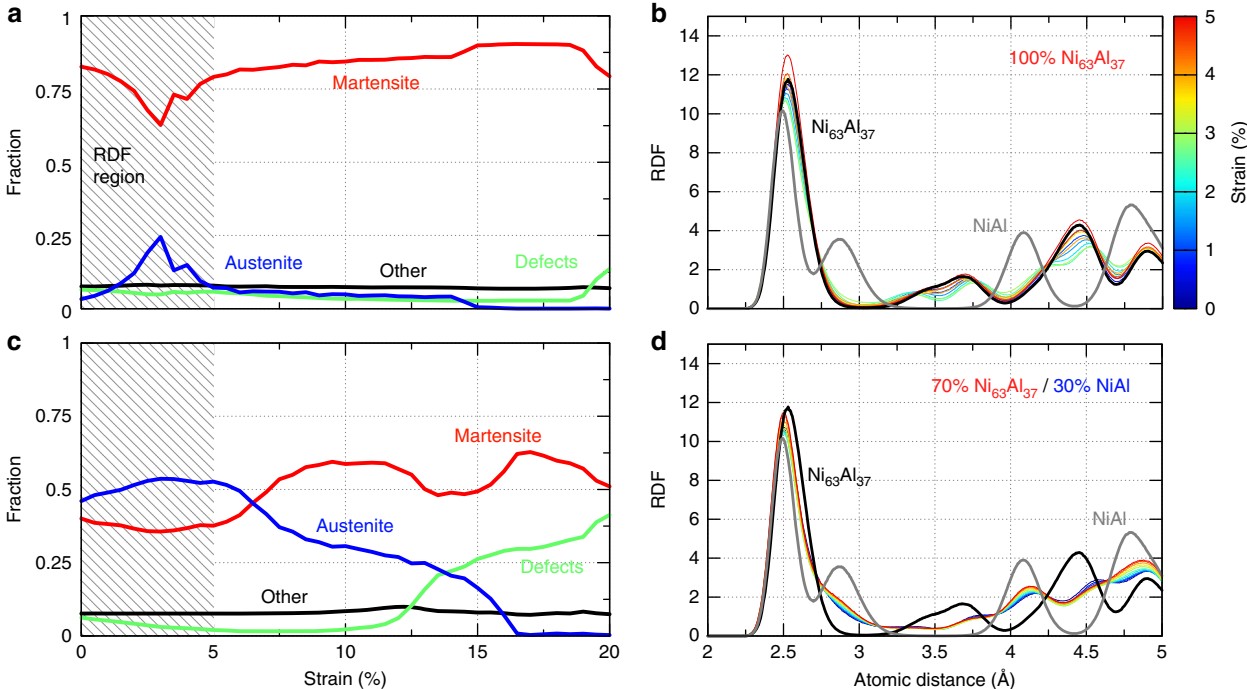

**Fig. 4** Structural analysis contrasting homogeneous and metamaterial nanowires. **a** CNA shows the homogeneous $Ni_{63}Al_{37}$ nanowire is almost entirely martensite through 20% uniaxial strain with a temporary, sharp increase in austenite during nucleation of a new domain orientation. **b** This results in moderate changes in the RDF up to 5% strain. Shading for CNA indicates region shown in atomic structures (Fig. 3) and RDF. RDF for unstrained homogeneous $Ni_{63}Al_{37}$ and NiAl nanowires in black and gray, respectively. **c** 70 at% T-Shell metamaterial nanowire CNA exhibits phase coexistence with smooth switching between martensite and austenite. **d** The metamaterial RDF lies between the individual phases, with significant peak broadening, which changes only slightly with strain

phase and shows superelastic behavior governed by a change in domain orientation. In order to develop an atomic-level picture of the processes of elastic and inelastic deformation, we analyzed the atomistic trajectories using a common neighbor analysis (CNA), as shown in Fig. 3, that enables the classification of atoms into phases and the identification of martensitic domains, see "Methods" section, Supplementary Methods, and Supplementary Table 1. Atoms belonging to the martensitic phase are colored in shades of red (dark red in $Ni_{63}Al_{37}$ and light red in NiAl), while atoms in an austenite environment are colored in shades of blue (dark blue in $Ni_{63}Al_{37}$ and light blue in NiAl). Stacking faults in the martensite (shown in green) can also be determined from the local bonding environment and are useful in identifying domains as they always lie on $\{110\}_{B2}$ planes that contain the short lattice parameter. The initial structure of the $Ni_{63}Al_{37}$ wire consists of a single martensitic domain (Fig. 3a) with its short lattice parameter, corresponding to $[001]_{B2}$, oriented along the wire axis with stacking faults on $(110)_{B2}$ and $(\overline{1}10)_{B2}$ planes. At ~2.5% strain, we observe the nucleation of a second, rotated martensitic variant with stacking faults along $(011)_{B2}$ and its short direction normal to the wire axis (along the $[100]_{B2}$). Thus, deformation is accommodated by domain switching, typical of shape memory alloys. It is interesting to note that domain switching involves the transient formation of austenite as an intermediate phase (see blue atoms in Fig. 3a and evolution of the atomic fractions of each phase in Fig. 4a). After nucleation, the new domain grows through domain wall motion to accommodate strain, requiring a rather low stress. When this mechanism is exhausted, the stress increases and deformation proceeds via the nucleation of the same unstable martensitic variant observed in the NiAl B2 case. The atomic trajectory during deformation is shown in Supplementary Movie 1. The significant stress required to nucleate and propagate domain switching is a key difference between

superelasticity and the ultra-low stiffness metamaterial response. The process of deformation for both components of the metamaterial are as expected, lending credence to the overall fidelity of the model to describe martensitic processes.

**Ultra-low stiffness with full strength.** Having established the response of the pure components to mechanical deformation, we switch our attention to that of the heteroepitaxial nanowire metamaterials. Typical stress-strain curves of Core($Ni_{63}Al_{37}$)/Shell(NiAl) (herein denoted T-Core for transforming core) and Core(NiAl)/Shell($Ni_{63}Al_{37}$) (denoted T-Shell) are shown in Fig. 2. These simulations confirm the expected, engineered, softening of the metamaterials, including ultra-low stiffness for the 70 at% T-Shell with a Young's modulus of 4 GPa, an order of magnitude smaller than that of the softest component. Atomistic trajectories, CNA, and radial distribution function (RDF) structural analysis for all metamaterial composite fractions in Fig. 2 are shown in Supplementary Figs. 1–4 and described in Supplementary Note 1. A detailed analysis of the atomistic structures provides key insight into the mechanisms responsible for the ultra-low stiffness. The CNA indicates that the undeformed 70 at% T-Shell metamaterial exhibits coexistence between the austenite phase and two martensitic domains, Fig. 3b and Fig. 4c. Note that the interfaces between the austenite and martensite phases are jagged with interpenetration between the two phases; furthermore, some regions of the $Ni_{63}Al_{37}$ shell are identified as austenite and some of the NiAl core as martensite (Fig. 3b), opposite their preference as isolated materials. The RDFs prove essential for understanding the intriguing CNA results. The metamaterial RDFs exhibit broad peaks, with features of both the B2 and martensite phases, Fig. 4d. Surprisingly, the RDFs of the metamaterial cannot be described as a linear combination of those of the austenite and martensitic

phase (Supplementary Fig. 5), indicating a range of local lattice parameters. These observations within the metamaterials—gradual transition between phases and variation in local lattice parameters across the sample—are clear manifestations of the interfacial stresses affecting the stability of the system. Moreover, this is consistent with the flat energy landscapes in Fig. 1, where a wide range of lattice parameters lie within a narrow energy range and is strikingly different from the pure $Ni_{63}Al_{37}$ sample. Application of a mechanical load to the 70 at% T-Shell wire results in the seamless change in local lattice parameter with a small change in free energy. For strain up to a few percent, tensile deformation results in the growth of the martensite variant oriented with its short direction normal to the axis of the wire while maintaining the continuous coexistence of both phases. The homogeneous $Ni_{63}Al_{37}$ wire contrasts this behavior, beginning and remaining in the martensitic phase (except the sharp increase in austenite fraction during the nucleation of the second, rotated domain), and showing narrower RDF peaks. Once the change in lattice parameter afforded by the flat region of the free energy landscape is exhausted, the 70 at% T-Shell wire stiffens and achieves an ultimate tensile strength comparable to the pure phases (see atomic trajectory in Supplementary Movie 2). Stress relaxation occurs via transformation to the same unstable martensite as in the pure $Ni_{63}Al_{37}$ and NiAl cases.

The ultra-low stiffness observed in the 70 at% T-Shell is both reversible and reproducible; furthermore, it applies to various compositions and geometries. We subjected a series of heteroepitaxial wires to cyclic mechanical deformation with a total strain range of 2%. Figure 5a shows the resulting stiffness of the nanowires as a function of the $Ni_{63}Al_{37}$ at%, contrasting it with the expectation from Voigt and Reuss standard composite averages. The range of Young's moduli indicates the variability resulting from cycling loading of multiple statistically independent samples, see "Methods" section, Supplementary Note 2, and Supplementary Fig. 6, where we show stress-strain curves for all samples. Remarkably, we obtain stiffness as low as ~2 GPa for a few individual samples (averaged over five deformation cycles); selected samples are shown in Supplementary Movies 3–6. The cyclic loading also highlights the lack of threshold stress, characteristic of superelastic materials, separating elastic deformation from a regime dominated by domain wall motion (see $Ni_{63}Al_{37}$ stress-strain response, Fig. 2). Further, since the deformation of the ultra-low stiffness metamaterials depends on coexistence of and transition among a wide range of local parameters, there is negligible hysteresis in the elastic response, another important differentiating factor from superelasticity. Once cycling above the elastic regime, we observe only slight hysteresis in the stress-strain response of the 70 at% T-Shell nanowire, Supplementary Fig. 7. The softening is not restricted to nanowire configurations; epitaxial laminates with a periodic thickness of 23.2 nm show similar softening, see inset in Fig. 5a and Supplementary Note 3.

The predicted properties of the epitaxial nanowires and laminates fill an important gap in materials properties not achievable with current classes of materials. This is displayed in an Ashby chart[31] of strength vs. modulus in Fig. 5b[32], where we compare standard experimental results to our metamaterials from MD simulations. This minimum Young's modulus is over an order of magnitude lower than that of the softer component and is unprecedented for a full density metal, either in experiment or any realistic simulation. We acknowledge that comparing the strength of defect-free nanostructures with bulk samples of the various classes of materials is not entirely fair; however, the combination of even moderate strength and ultra-low stiffness for a metal is otherwise unfilled. Significant tuning of stiffness is possible in cell structures and via macroscopic structural design.

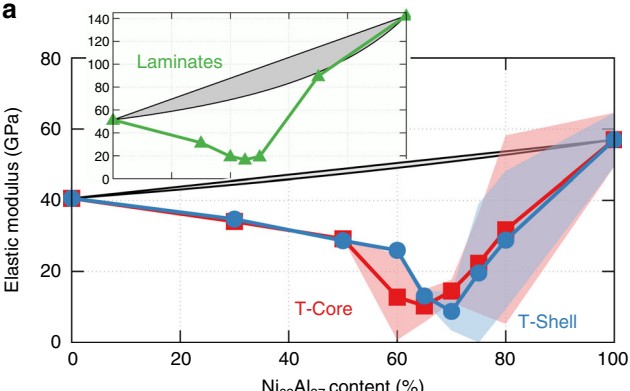

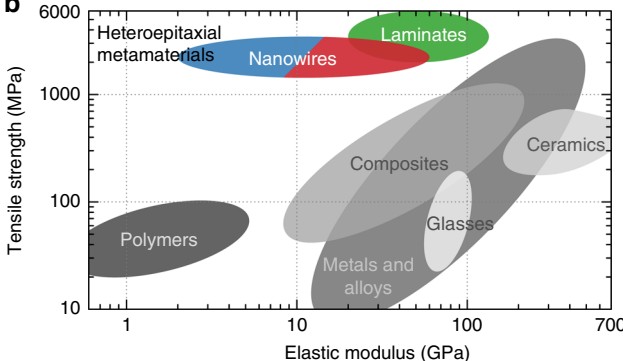

**Fig. 5** Ultra-low stiffness for nanoscale heteroepitaxial metamaterials. **a** Stiffness of metamaterial nanowires as a function of $Ni_{63}Al_{37}$ composite fraction for T-Core and T-Shell configurations. Black lines (and gray shading) show standard rule of mixtures. Shaded region shows ±2 standard deviations from cyclic loading of multiple independent samples with randomized compositions (see "Methods" section, Supplementary Note 2, and Supplementary Fig. 6). Inset shows stiffness of nanolaminate structures with identical x axis (Supplementary Note 4). **b** Ashby plot showing predicted properties for our epitaxial metamaterials in comparison to standard metals, polymers, ceramics, and composites. Materials data from CES EduPack 2015, Granta Design Limited, Cambridge, UK, 2015

Even negative stiffness has been demonstrated under quasi-static, displacement-controlled conditions in systems that incorporate buckled elements[33]. In contrast to our work, this is achieved at the structural, not material, level and at the expense of strength.

## Discussion

As with any prediction in computational materials science, two key questions need to be addressed: whether the results are robust with respect to the assumptions in the model and whether the proposed systems are realizable experimentally. The next paragraphs address these key issues.

To assess whether the softening was strongly dependent on the model used to describe the alloy, we repeated the simulations with a second, independently developed interatomic potential[34]. While this second potential was parameterized using different data and exhibits quite different behavior in terms of the energy landscape, elastic properties, and martensitic transformation, the resulting metamaterials also exhibit significant softening relative to the expected composite averages, see Supplementary Figs. 8 and 9 and Supplementary Note 4. We observe a reduction in the stiffness of 67% compared with the softest component; the slightly lower degree of softening as compared to the results shown in Fig. 5 is due to a less ideal energy landscape combination, see Supplementary Fig. 8. We note that neither interatomic potentials

were modified or tuned in any way for this paper. As discussed above, we also explored heteroepitaxial laminated metamaterials and found them to exhibit similar trends (inset in Fig. 5a and Supplementary Note 3). We therefore conclude that the ultra-low stiffness observed is not an artifact of the models used; rather, the unique properties of the metamaterials results from the general features of the energy landscapes underlying the two components of the metamaterial.

The possibility of synthesizing structures similar to those studied here and the challenge of epitaxial integration also deserves discussion. Coherent core/shell nanowires[35], nanolaminates, and other nanostructures[36] have been demonstrated in semiconductors and oxides. While heteroepitaxy in metals is more challenging, coherent superlattices have been demonstrated in various metallic systems via magnetron sputtering[37] and molecular beam epitaxy[38]. The lattice mismatch between components could constrain the maximum thickness of epitaxial layers[39], yet the fact that one of the materials exhibits a martensitic transformation with a concave (negative stiffness) region of free energy pushes the coherence limit to significantly larger lengths. In fact, Buschbeck et al.[40] demonstrated the coherent integration of Fe–Pd magnetic shape memory alloys with a family of substrates spanning the entire Bain path. A process that could affect the properties of the metamaterials is interlayer diffusion. To assess the sensitivity of our results to this process, T-Shell nanowires were created with a graded composition across the interface, see Supplementary Note 5. These wires have 65 at% Ni in the shell and the Ni concentration varies from 65 at% at a radius of 45 nm to 50 at% at 30 nm, see Supplementary Fig. 10a. The stiffness of these nanowires under cyclic loading fall between 7 and 15 GPa, Supplementary Fig. 10b, within the range of stiffness of the sharp interface T-Shell wires with the same overall composition (70 at%) shown in Fig. 5a.

Finally, we note that our approach does not require core/shell or laminate geometries; coherent precipitates in a matrix are expected to exhibit the same effect. Interestingly, coherent, nanoscale $Ti_2Cu$ precipitates formed during heat treatment of TiNiCu shape memory alloys have been reported[41]. The authors demonstrated orientation relationships between the matrix and precipitates during multiple martensite–austenite transformation cycles, significantly improving the fatigue life of the alloy.

In summary, by harnessing the negative stiffness state of a martensitic material and stabilizing it at the nanoscale via epitaxial integration, we demonstrated an ultra-low stiffness, full density metal. The metamaterial was designed by identifying an appropriate second component capable of stabilizing the negative stiffness region. Large-scale MD simulations show that the flat energy landscape underlying the metamaterial does, in fact, result in ultra-low stiffness. The metamaterial nanowires exhibit Young's moduli in the low gigapascal regime, similar to that of many polymers, as a wide range of lattice parameters result in nearly identical energies. This softening is achieved at full density and with no loss of strength.

While demonstrated here for model NiAl alloys, the use of negative stiffness phases and landscape engineering to achieve mechanical properties not otherwise possible is quite general and applicable to any material exhibiting martensitic transformations, including ferroelectric[19, 20, 22, 42, 43] materials. The results in refs. [20, 22] and related work also harness negative stiffness states; however, in contrast to our work, their systems are not epitaxial and their properties are transient. The authors use the negative stiffness portion of the energy landscape of ferroelectric inclusions, constrained in a matrix, during thermally induced transformation to produce unique properties. Ultra-high dynamic stiffness and viscoelastic damping result from negative compressibility and shear modulus of the inclusions, respectively. Our metamaterials stabilize the negative stiffness region of one

phase by its coherent integration with a stable phase at the nanoscale. Not only does this produce ultra-low stiffness due to the stabilization of a wide range of local lattice parameters, but this is accomplished under quasi-static, rather than dynamic, conditions.

We further note that free energy landscape engineering can be used for purposes other than achieving ultra-low stiffness; for example, the technique can be used to modify martensitic transition temperatures and thermal hysteresis[44]. Metals with stiffness in the low gigapascal range are highly desirable for integration with soft materials with similar stiffnesses. For example, the combination of ultra-low stiffness with metallic electronic conductivity and high strength furthers the possibilities for multifunctional, flexible electronics[45] and would be desirable for implantable devices, enabling better integration with tissue[46, 47].

## Methods

**MD simulation details**. All MD simulations were performed with the LAMMPS code[48] with atomic visualization and CNA done using OVITO[49, 50]. Within LAMMPS, a velocity-Verlet algorithm integrated with a 1 fs time step was used throughout. All simulations used a Nose–Hoover thermostat with a coupling constant of 0.01 ps; those with pressure control used a Nose–Hoover barostat with a coupling constant of 0.1 ps.

**Interatomic model**. The interatomic model from Farkas et al.[30] was fit to experimental properties of Ni, Al, and $Ni_3Al$ from Voter et al.[51], B2 NiAl[52], as well as 3R $L1_0$ martensite and $Ni_5Al_3$[29]. It is able to capture the main details of martensitic transformation in NiAl: the composition range for transformation and approximate transformation temperatures, in addition to the relative stability of the austenite and martensite. However, the potential does not predict exactly the complex R7 martensitic phase seen experimentally.

**Composition randomization**. The off-stoichiometric structures were built by first generating equiatomic B2 NiAl systems of desired dimensions. The disordered $Ni_{63}Al_{37}$ phase was then introduced by randomly switching Al atoms for Ni atoms until the 63% ratio was reached in the chosen region; see additional discussion in ref. [29].

**MD free energy calculations**. The free energy landscapes were generated by straining samples of each single-phase $Ni_{50}Al_{50}$ and $Ni_{63}Al_{37}$ at 300 K with an NVT (isothermal, isochoric) ensemble. The systems were initialized as perfect B2 crystals of 512,000 atoms with {100} orientation (fully periodic boundaries), equilibrated in all directions, and then continuously strained biaxially between 2.7 and 3.1 Å at a strain rate of $7 \times 10^8 \, \text{s}^{-1}$. The third axis was left free to equilibrate during deformation with 1 atm pressure, as were all angles. The total free energy change for this isothermal process can be computed by $dF = V(\sigma_{xx}\epsilon_{xx} + \sigma_{yy}\epsilon_{yy})$ and numerically integrated[44]. The metamaterial landscapes were produced by combining the pure NiAl and $Ni_{63}Al_{37}$ landscapes in varying fractions.

**MD mechanical testing**. The metamaterial nanowires contained 787,400 atoms, built in the {100} orientation with periodic boundary conditions along the wire axis, initially measuring 14.5 by 58.1 nm in diameter and length, respectively. Each wire was thermalized at 300 K for 100 ps prior to deformation. Full strain mechanical tests consisted of uniaxial deformation along the wire axis to 20% engineering strain for 400 ps (strain rate of $5 \times 10^8 \, \text{s}^{-1}$) under an NVT ensemble. The stiffness for each composite fraction was obtained using multiple samples, each with unique randomizations for composition and atomic velocities. Samples were equilibrated for 100 ps, followed by two sets of alternating single period sine-wave strain and 50 ps equilibration. Five sine-wave strain cycles were then run, used for averaging and variability of the stiffness, taking linear fits every 0.25% in 1% increments. Each cycle was between 0 and 2% strain at a rate of $5 \times 10^8 \, \text{s}^{-1}$. For fractions above 50 at% $Ni_{63}Al_{37}$, 10 samples were used due to significant variability from differing domain formation; at and below 50 at%, two samples were found to be sufficient to describe the uncertainty (Supplementary Note 2).

**Data availability**. Thermodynamic and selected structural data, as well as scripts for plotting and analyzing the data are available from the Strachan Research Group repository (https://github.rcac.purdue.edu/StrachanGroup/ultralow-stiffness-NiAl). The atomistic data from this study are available from the corresponding author upon reasonable request. Direct simulations for reproducibility are available through nanoHUB.org with the Nanomaterial Mechanics Explorer simulation tool[53] (https://nanohub.org/tools/nanomatmech).

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

## Acknowledgements

This work was supported by the United States Department of Energy Basic Energy Sciences (DoE-BES) program under program no. DE-FG02-07ER46399 (program manager J. Vetrano). Computational resources from nanoHUB and Purdue University are gratefully acknowledged.

## Author contributions

A.S. planned and supervised the research. S.T.R. and A.B.-R. carried out the simulations. All authors contributed to discussion and analysis of the data and prepared the manuscript.

## Additional information

**Competing interests:** The authors declare no competing financial interests.

