## [Peer Review File · Nature Communications]

Reviewers' Comments:

Reviewer #1:

Remarks to the Author:

Review: S. Reeve, A. Belessiotis-Richards & A. Strachan, " Ultra-low stiffness metals: Harnessing mechanical instabilities at the nanoscale." By computational modeling the authors show that a stress-strain behavior with a low tangent modulus can be achieved by enabling a controlled displacive phase transformation in a material. Through investigation of the free energy of epitaxially grown nickel-aluminium alloys with specific stoichiometry, the investigators find a compound of 2 nickel aluminoids that shows a very low change in energy as its lattice parameter is varied. The implication is that the tangent modulus of such a material will be very low, with the strain imposed being tantamount to controlling the displacive phase transformation that evolves the alloy. This result is both very insightful and not surprising, though the authors are to be congratulated for finding the conditions that make a reality of the behavior they have identified. The reason why I say that the approach is very insightful is because the authors have had the wit to implement an innovative approach for finding a highly useful set of properties for a material that will enable exploitation in a number of ways that the authors themselves point out. At the same time, the reason I say that the behavior is not surprising is that equivalent conditions have been known to occur in other materials, but in a differing regimes. I refer specifically to ferroelectrics, where, in materials such as PZT, the very high dielectric constant is due to domain wall motion that converts one variant to another where the domains have radically different polarization. This process gives a much bigger effect than can be achieved through the intrinsic electronic polarization within atoms that is the usual source of material polarization and that gives rise to relatively low dielectric constants. Furthermore, in superelastic shape memory alloys the same behavior is observed as identified by the authors, with conversion of austenite to martensite and vice versa upon straining and unstraining. The difference is that in NiTi superelastic materials there is a distinct stiff elastic response of austenite first before the low tangent modulus segment of the stress-strain curve develops, plus there is hysteresis upon unloading. In my view the authors should discuss this behavior of superelastic shape memory alloys to put their discoveries in context. In particular I am interested in whether there is a significant hysteresis upon unloading in the material that the authors have investigated. I think a brief comment on materials such as PZT would also be useful. While I am carping in regard to the uniqueness of the authors' result, I nevertheless praise them for finding the conditions that enable a very low tangent modulus in the stress-strain behavior of the material they have devised. In my view the authors spread confusion when they talk about rules of mixtures and bounds on elastic properties. The authors are not implementing elasticity in their simulations and therefore it is misleading to discuss the results in the context of concepts that apply strictly to elastic conditions. The authors should clear up this danger of confusion by making it clear they are mixing apples and oranges when they describe their results in terms of rules of mixtures and bounds on elastic properties. The authors should separate more clearly their descriptions of results that are computational and those that are experimental. Often the computational results are described as if they were obtained experimentally, and this will mislead readers in regard to what is practical and what is virtual in their results. I have the suspicion that the material that the authors have invented will experience a formation of domains of variants having contrasting lattice parameter values, somewhat equivalent to the existence of ferroelectric domains in materials such as PZT. In my view the authors should address this point by discussing whether such domains are likely or not likely to form in their material.

Reviewer #2:

Remarks to the Author:

This work uses molecular dynamics simulations to show that two materials with different lattice constants can be combined to create a composite with an ultra-low stiffness. This work is good enough for publication in a more specialized journal, but is not suitable for Nature Communications for the following specific reasons:

1. Materials with two stable/metastable phases exhibit a negative stiffness. When such a material is combined with another material with a positive stiffness, the stiffness of the composite can be reduced. While the idea is interesting, the work to prove it is not to just use MD simulations to show the reduced stiffness (which will be almost certainly true because we are manually constructing an idealized coherent model to reproduce the idea). Additionally, the authors should do more simulations to prove such composites are stable and can be manufactured in reality. How the stability changes with misfit dislocations at the interfaces, how misfit dislocation density changes the predicted stiffness, if there are interlayer diffusion between NiAl and Ni₆₃Al₃₇, will the interfacial energy a challenge to overcome, etc., are just example questions that need to be answered for publication in Nature Communications.
2. Material with a low stiffness is sought because it can be mechanical more compatible with other materials. In some ways, the present work does not solve the problem, but rather transfer the problem. Yes, the compatibility problem between this and another material is reduced, but the compatibility problem within this material is increased because now this material is composed of two materials with different stiffness values. The work does not provide any analysis of the latter compatibility.
3. Experimental validation can be powerful to make a work publishable in Nature Communications. That does not exclude pure modeling works. However, more careful modeling efforts are needed if no experimental validation data is currently available. Will the errors of the interatomic potential change the conclusion? Although authors have tried another potential, but more detailed analysis as why and why not should be given. What if the lattice constant of the energy minimum of one material does not match the lattice constant of the energy maximum of another material? This also relates to my comment #1 as if all possibilities of model errors are considered.

Reviewer #3:

Remarks to the Author:

The manuscript reports a Molecular Dynamics investigation into the development of metal alloy nanowires and laminates displaying ultra-low stiffness by virtue of incorporating a negative stiffness within the composite metal system. The specific system requires the unstable state associated with the martensitic transformation in Ni-rich Ni_xAl_{1-x} to be stabilised through epitaxial integration with NiAl. The simulations appear to have been performed with care and appropriate attention to detail, including testing for an alternative interatomic potential to that used for the main body of work. Significant reduction in stiffness of the system is achieved, with potential applications noted in the discussion.

The idea of incorporating and stabilising negative stiffness components within a two-phase composite system is not new and the authors acknowledge this through reference to the work of Lakes and co-workers (refs [18] and [19]). The authors make clear the work reported in this paper is distinct from these references in so much as the reported properties here are quasi-static rather than transient dynamic properties in [18, 19]. Nevertheless, given that [19] reports (dynamic) stiffness greater than diamond (and greater than either of the constituent components of the composite), the first sentence on page 3 which states "we use theory to design the first composite that defies the constraints of the rule of mixtures for a property insensitive to microstructure: stiffness" needs amending since it is clearly not the first example of a composite exceeding the

bounds for stiffness.

In addition to the exceeding bounds claim in the aforementioned sentence, the insensitivity of stiffness on microstructure needs qualifying since cellular solids are well known to show very clear dependence of stiffness on cellular microstructure (see, for example, L. J. Gibson and M. F. Ashby, *Cellular Solids: Structure and Properties*, Cambridge University Press, Cambridge, 1999).

The incorporation of negative stiffness components within a 2nd component in a composite system that then demonstrates very low (and even, under displacement control, negative) quasi-static stiffness has recently been reported both theoretically and experimentally by Hewage and co-workers (T. A. M. Hewage, K. L. Alderson, A. Alderson and F. Scarpa, *Adv. Mater.* 2016, 28, 10323–10332). The system reported by the authors is distinct from that of Hewage in so much as in this paper nanocomposites are considered whereas Hewage and co-workers report a much larger scale system. Nevertheless, the existence of the recent report of a quasi-static low stiffness composite containing negative stiffness inclusions should be acknowledged in this paper.

The present paper focusses on ultra-low stiffness. Given the ultra-high stiffness due to negative stiffness inclusions reported in [19], the authors are invited to briefly discuss the use of negative stiffness phases in achieving both extremes in stiffness and to compare and contrast their system with that in [19] in this respect.

Finally, the authors are invited to review their use of the term 'metamaterial' in several places to describe their composite systems. This term usually implies a system deriving unusual effective properties from a regular arrangement of 'sub-units' or 'building blocks', and is most usually applied to electromagnetic properties. So justification and reference to an appropriate definition in the literature is needed to qualify the use of this term, or remove and replace with a more appropriate term.

On balance, whilst the incorporation of negative stiffness within a 2-phase composite to achieve low stiffness (including quasi-static) is not new, making this a marginal assessment, the embodiment of this in nanowires and nanolaminates probably warrants publication of this manuscript in *Nature Communications* once the authors have addressed the issues raised above.

Reply to Reviewer reports

We would like to thank the Reviewers for their careful analysis of our work and their insightful comments. Please find our response to each comment below.

Revisions pertaining to comments from Reviewer #1 are underlined in green, from Reviewer #2 underlined in orange, and from Reviewer #3 underlined in purple in the revised manuscript.

Reviewer #1 (Remarks to the Author):

Reviewer comment:

By computational modeling the authors show that a stress-strain behavior with a low tangent modulus can be achieved by enabling a controlled displacive phase transformation in a material. Through investigation of the free energy of epitaxially grown nickel-aluminium alloys with specific stoichiometry, the investigators find a compound of 2 nickel aluminoids that shows a very low change in energy as its lattice parameter is varied. The implication is that the tangent modulus of such a material will be very low, with the strain imposed being tantamount to controlling the displacive phase transformation that evolves the alloy.

This result is both very insightful and not surprising, though the authors are to be congratulated for finding the conditions that make a reality of the behavior they have identified. The reason why I say that the approach is very insightful is because the authors have had the wit to implement an innovative approach for finding a highly useful set of properties for a material that will enable exploitation in a number of ways that the authors themselves point out. At the same time, the reason I say that the behavior is not surprising is that equivalent conditions have been known to occur in other materials, but in a differing regimes. I refer specifically to ferroelectrics, where, in materials such as PZT, the very high dielectric constant is due to domain wall motion that converts one variant to another where the domains have radically different polarization. This process gives a much bigger effect than can be achieved through the intrinsic electronic polarization within atoms that is the usual source of material polarization and that gives rise to relatively low dielectric constants. Furthermore, in superelastic shape memory alloys the same behavior is observed as identified by the authors, with conversion of austenite to martensite and vice versa upon straining and unstraining. The difference is that in NiTi superelastic materials there is a distinct stiff elastic response of austenite first before the low tangent modulus segment of the stress-strain curve develops, plus there is hysteresis upon unloading. In my view the authors should discuss this behavior of superelastic shape memory alloys to put their discoveries in context.

Authors response:

Thanks for this comment. We agree with the Reviewer that there are similarities between the mechanisms behind the ultra-low stiffness of our metamaterial and shape memory or ferroelectric domain wall motion, but there is also an important distinction. Inducing an austenitic/martensitic transition or domain wall motion (in superelasticity and ferroelectrics) requires a threshold stress and the materials respond elastically (with standard values of stiffness) until this threshold is achieved. This is shown in the pure

Ni₆₃Al₃₇ case in our paper, see Figure 2. While moving domains often require relatively low stresses, these materials exhibit an elastic regime with standard values of stiffness. Similarly, moving domain walls and converting polarization variants in ferroelectrics lead to very high dielectric constant during transformation. As the Reviewer points out, this is well documented. Our materials are novel in that both phases and a range of intermediate lattice parameters have nearly identical stability. Thus, a structural analysis of the ultra-low stiffness materials shows austenite and martensite phases interpenetrating each other and the deformation of the wires and laminates involves very low stresses without any threshold stress required. This distinction is discussed in more detail in Section 4 of the revised version. See our answer to the next, related, question.

Reviewer comment:

In particular I am interested in whether there is a significant hysteresis upon unloading in the material that the authors have investigated. I think a brief comment on materials such as PZT would also be useful.

Authors response:

We observe almost no hysteresis in the elastic cycling stress-strain curves for the lowest stiffness cases featured in the main paper. Higher composite fraction nanowires do show hysteresis. See Figure S6 in the Supplementary Information. This is an important point and is discussed in more detail in Section 4 and Supplementary Section 4 of the revised version of the paper:

“The cyclic loading also highlights the lack of threshold stress, characteristic of superelastic materials, separating elastic deformation from a regime dominated by domain wall motion (see Ni₆₃Al₃₇ stress-strain response, Fig. 2). Further, since the deformation of the ultra-low stiffness metamaterials depends on coexistence of and transition among a wide range of local parameters there is negligible hysteresis in the elastic response, another important differentiating factor from superelasticity.”

Reviewer comment:

While I am carping in regard to the uniqueness of the authors’ result, I nevertheless praise them for finding the conditions that enable a very low tangent modulus in the stress-strain behavior of the material they have devised. In my view the authors spread confusion when they talk about rules of mixtures and bounds on elastic properties. The authors are not implementing elasticity in their simulations and therefore it is misleading to discuss the results in the context of concepts that apply strictly to elastic conditions. The authors should clear up this danger of confusion by making it clear they are mixing apples and oranges when they describe their results in terms of rules of mixtures and bounds on elastic properties. The authors should separate more clearly their descriptions of results that are computational and those that are experimental. Often the computational results are described as if they were obtained experimentally, and this will mislead readers in regard to what is practical and what is virtual in their results.

Authors response:

Let us start with the first part of the comment. We use molecular dynamics simulations which describes elasticity explicitly; of course, MD also describes processes beyond elasticity including phase transformations and plasticity. What is more, molecular dynamics simulations are well known to result in

elastic constants that follow the rules of mixtures in composites. See the following for examples of elastic properties from molecular dynamics: Rassoulinejad-Mousavi, S. M., Mao, Y. & Zhang, Y. Evaluation of copper, aluminum, and nickel interatomic potentials on predicting the elastic properties. *J. Appl. Phys.* **119**, (2016).

Regarding the second part of the comment, we agree with the Referee. The revised version of the paper stresses the origin of the results presented to avoid confusion:

“we compare standard experimental results to our metamaterials from MD simulations.”

Reviewer comment:

I have the suspicion that the material that the authors have invented will experience a formation of domains of variants having contrasting lattice parameter values, somewhat equivalent to the existence of ferroelectric domains in materials such as PZT. In my view the authors should address this point by discussing whether such domains are likely or not likely to form in their material.

Authors response:

This is a key point of our approach. Multiple domain structures appear when distinct domains have equivalent energies and are separated by an energy barrier, see the energy landscape for the $\text{Ni}_{63}\text{Al}_{37}$ system in Figure 1 (red curve). However, the metamaterial is designed so that a relatively wide range of lattice parameters exhibit nearly identical energies, see the energy landscape colored in black. A structural analysis shows the coexistence of a wide distribution of bond distances (Fig. 4) and interpenetrating austenite and martensite regions (Fig. 3).

Reviewer #2 (Remarks to the Author):

Reviewer comment:

This work uses molecular dynamics simulations to show that two materials with different lattice constants can be combined to create a composite with an ultra-low stiffness. This work is good enough for publication in a more specialized journal, but is not suitable for Nature Communications for the following specific reasons:

1. Materials with two stable/metastable phases exhibit a negative stiffness. When such a material is combined with another material with a positive stiffness, the stiffness of the composite can be reduced. While the idea is interesting, the work to prove it is not to just use MD simulations to show the reduced stiffness (which will be almost certainly true because we are manually constructing an idealized coherent model to reproduce the idea). Additionally, the authors should do more simulations to prove such composites are stable and can be manufactured in reality. How the stability changes with misfit dislocations at the interfaces, how misfit dislocation density changes the predicted stiffness, if there are interlayer diffusion between NiAl and Ni₆₃Al₃₇, will the interfacial energy a challenge to overcome, etc., are just example questions that need to be answered for publication in Nature Communications.

Authors response:

We thank the Reviewer for this important comment. While we do not minimize the challenges involved in realizing the proposed materials experimentally, coherent metallic superlattices, with negligible misfit defects, have been reported by several groups; see Refs. [36-37] in our manuscript. In addition, materials that exhibit martensitic transformations are known to exhibit large coherence limits and to be simpler to integrate epitaxially even under larger lattice parameter mismatch, see Ref. [39]. For these reasons, we believe that that the proposed structures are possible and that investigating the role of misfit dislocations, while interesting, is not critical at this point.

Interlayer diffusion is possible and even likely in the systems studied. Thus, following the Reviewer's suggestion, we performed additional simulations and showed that atomic interlayer diffusion does not affect the stiffness of the metamaterials in an appreciable manner. These new results are discussed in Section 5 of the revised version of the manuscript and its details included in the Supplementary Information, Section S5.

“A process that could affect the properties of the metamaterials is interlayer diffusion. To assess the sensitivity of our results to this process, T-Shell nanowires were created with a graded composition across the interface, see Supplementary Section 5. These wires have a 65 at. % in the shell and the Ni concentration varies from 65 at. % at radius of 45 nm to 50 at. % at 30 nm, see Supplementary Figure S8a. The stiffness of these new nanowires under cyclic loading fall between 7 – 15 GPa, Figure Supplementary S8b, within the range of stiffness of the sharp interface T-Shell wires with the same overall composition (70 at. %) shown in Fig. 5a.”

S5 Diffuse interface nanowires

“To establish that the ultra-low stiffness results are insensitive to diffusion between components, we created 60-70 at. % T-Shell nanowires with a diffuse compositional interface. The shell region was built as described in the Methods; within a distance of three unit cells

radially nearest the interface, the target composition was a linear interpolation between 0 and 13% added (randomized) Ni. Figure S8a compares the fraction of randomized nickel as a function of nanowire radius. The example from the main paper shows the abrupt change from the NiAl to Ni₆₃Al₃₇, contrasted by the diffuse systems. For all structures there is noise in the randomized section on a per layer basis, though the overall composition in the core and shell regions average to the prescribed fraction. Due to the added randomized composition for the graded interfaces, each is compared to T-Shell sharp interface systems with 5% larger shell regions. The elastic response is shown in Figure S8b. When comparing the correct phase fraction samples, the results for ten samples with independently randomized composition for each T-Shell fraction fall almost entirely within the ranges of the sharp interface samples from the main paper. For a minority of cases of graded interface 65 (effective) at. % T-Shell nanowires fall outside the sharp interface range as the specific distribution of randomized nickel prefers a single phase structure, as in the sharp interface 60 at. % T-Shell.”

Given the following facts:

- i) coherent metallic heterostructures have been experimentally created in the past,
- ii) interlayer diffusion does not affect our main results as demonstrated by our new simulations,
- iii) the results are obtained for two independent interatomic potentials and two different geometric arrangements,
- iv) the approach is not restricted to the NiAl system we studied,

We believe that the experimental realization of low-stiffness metals using our approach is likely and the theoretical/modeling effort in this paper provides the foundation for this future work.

Reviewer comment:

2. Material with a low stiffness is sought because it can be mechanical more compatible with other materials. In some ways, the present work does not solve the problem, but rather transfer the problem. Yes, the compatibility problem between this and another material is reduced, but the compatibility problem within this material is increased because now this material is composed of two materials with different stiffness values. The work does not provide any analysis of the latter compatibility.

Authors response:

We thank the Reviewer for bringing this up. Actually, since the proposed nanostructured metamaterials respond to mechanical deformation as a whole (both components deform together) there are effectively no internal interfaces. This is true for quasistatic loading, as described in our paper. Understanding the propagation of acoustic waves and shocks in these materials would be of interest but beyond the scope of the current paper.

Reviewer comment:

3. Experimental validation can be powerful to make a work publishable in Nature Communications. That does not exclude pure modeling works. However, more careful modeling efforts are needed if no experimental validation data is currently available. Will the errors of the interatomic potential change the conclusion? Although authors have tried another potential, but more detailed analysis as why and why

not should be given. What if the lattice constant of the energy minimum of one material does not match the lattice constant of the energy maximum of another material? This also relates to my comment #1 as if all possibilities of model errors are considered.

Authors response:

We agree that experimental confirmation of these predictions will be important and we are starting to work along this direction with collaborators. Yet, as the Reviewer points out, pure modeling work has its place. The key question to address when direct experimental validation is unavailable is whether the conclusions drawn are robust with respect to uncertainties and assumptions in the model. In the case of the current manuscript, several facts support the robustness of the results:

- i) The results of the explicit MD simulations are consistent with the expectation based on the engineered energy landscapes and based on sound, relatively simple, physics.
- ii) Two independent interatomic potentials yield the same trends.
- iii) The low stiffness is observed in nanowire and laminate setups as well as in the presence of diffuse interfaces.

The Reviewer is correct in pointing out that if the energy landscapes of the two components were very different the approach would not work. In the case of the system selected, the interatomic potentials predict the experimental lattice parameters with reasonable accuracy, see Ref. [29], such that the energy landscape overlap should be similar. In addition, we stress that the proposed method may be experimentally realizable in systems other than NiAl.

Reviewer #3 (Remarks to the Author):

Reviewer comment:

The manuscript reports a Molecular Dynamics investigation into the development of metal alloy nanowires and laminates displaying ultra-low stiffness by virtue of incorporating a negative stiffness within the composite metal system. The specific system requires the unstable state associated with the martensitic transformation in Ni-rich $\text{Ni}_{1-x}\text{Al}_x$ to be stabilised through epitaxial integration with NiAl. The simulations appear to have been performed with care and appropriate attention to detail, including testing for an alternative interatomic potential to that used for the main body of work. Significant reduction in stiffness of the system is achieved, with potential applications noted in the discussion.

The idea of incorporating and stabilising negative stiffness components within a two-phase composite system is not new and the authors acknowledge this through reference to the work of Lakes and co-workers (refs [18] and [19]). The authors make clear the work reported in this paper is distinct from these references in so much as the reported properties here are quasi-static rather than transient dynamic properties in [18, 19]. Nevertheless, given that [19] reports (dynamic) stiffness greater than diamond (and greater than either of the constituent components of the composite), the first sentence on page 3 which states “we use theory to design the first composite that defies the constraints of the rule of mixtures for a property insensitive to microstructure: stiffness” needs amending since it is clearly not the first example of a composite exceeding the bounds for stiffness.

In addition to the exceeding bounds claim in the aforementioned sentence, the insensitivity of stiffness on microstructure needs qualifying since cellular solids are well known to show very clear dependence of stiffness on cellular microstructure (see, for example, L. J. Gibson and M. F. Ashby, Cellular Solids: Structure and Properties, Cambridge University Press, Cambridge, 1999).

Authors response:

We agree with the Referee and certainly did not try to overstate our results. We have modified the statement to:

“we use theory to design the first composite that defies the constraints of the rule of mixtures for a **quasistatic** property **weakly sensitive** to microstructure: stiffness.”

With regards to the last point, we have added text to make clear the distinction between microstructural effects and materials such as cellular solids:

“We note that stiffness depends on microstructure (via texture), but to a much lesser degree than strength and toughness, and can to date only be engineered to a large extent via open structures such as foams and lattices, which incorporate air as a second phase.”

Reviewer comment:

The incorporation of negative stiffness components within a 2nd component in a composite system that then demonstrates very low (and even, under displacement control, negative) quasi-static stiffness has recently been reported both theoretically and experimentally by Hewage and co-workers (T. A. M. Hewage, K. L. Alderson, A. Alderson and F. Scarpa, Adv. Mater. 2016, 28, 10323–10332). The system reported by the authors is distinct from that of Hewage in so much as in this paper nanocomposites are

considered whereas Hewage and co-workers report a much larger scale system. Nevertheless, the existence of the recent report of a quasi-static low stiffness composite containing negative stiffness inclusions should be acknowledged in this paper.

Authors response:

We thank the Reviewer for bringing this related work to our attention. We agree with the similarities and differences highlighted. An additional important distinction between our work and that of Hewage et al. is the distinct lack of threshold stress for our materials. We have cited Hewage et al. and added discussion within Section 1 of the manuscript:

“Significant tuning of stiffness is possible in cell structures and via macroscopic structural design. Even negative stiffness has been demonstrated under quasistatic, displacement-controlled conditions in systems that incorporate buckled elements³². In contrast to our work, this is achieved at the structural, not material, level and at the expense of strength.”

Reviewer comment

The present paper focusses on ultra-low stiffness. Given the ultra-high stiffness due to negative stiffness inclusions reported in [19], the authors are invited to briefly discuss the use of negative stiffness phases in achieving both extremes in stiffness and to compare and contrast their system with that in [19] in this respect.

Authors response:

We have added additional discussion to clarify the similarities and differences to Lakes et al. in Section 6:

“The results in Refs. [19,21] and related work also harness negative stiffness states; however, in contrast to our work, their systems are not epitaxial and their properties are transient. The authors use the negative stiffness portion of the energy landscape for ferroelectric inclusions, constrained in a matrix, during thermally induced transformation to produce unique properties. Ultra-high dynamic stiffness and viscoelastic damping result from negative compressibility and shear modulus of the inclusions, respectively. Our metamaterials stabilize the negative stiffness region of one phase by its coherent integration with a stable phase at the nanoscale. Not only does this produce ultra-low stiffness due to the stabilization of a wide range of local lattice parameters, but this is accomplished under quasi-static, rather than dynamic, conditions.”

Reviewer comment

Finally, the authors are invited to review their use of the term ‘metamaterial’ in several places to describe their composite systems. This term usually implies a system deriving unusual effective properties from a regular arrangement of ‘sub-units’ or ‘building blocks’, and is most usually applied to electromagnetic properties. So justification and reference to an appropriate definition in the literature is needed to qualify the use of this term, or remove and replace with a more appropriate term.

Authors response:

While metamaterial is most often applied to electromagnetic properties, we believe it is the best term to describe our composites. We now cite a review of metatmaterials (Ref. [17]): Kadic et al. Metamaterials

beyond electromagnetism. *Rep. Prog. Phys.* 76, 126501 (2013). The structural part of their definition states:

“Metamaterials are rationally designed man-made structures composed of functional building blocks”

They go on to describe the effect of structure:

“to realize extreme or even unheard of effective material properties that go quantitatively and qualitatively beyond (meta, Greek) what is available in natural substances”

Where both portions describe the design and results for our materials. Even definitions from reviews discussing exclusively electromagnetic metamaterials (Soukoulis, C. M. & Wegener, M. Past achievements and future challenges in the development of three-dimensional photonic metamaterials. *Nat. Photonics* 52, 47–50 (2011).) can easily describe mechanical systems:

“*Photonic* metamaterials are man-made structures composed of tailored micro- or nanostructured *metallodielectric subwave-length* building blocks. This deceptively simple yet powerful concept allows the realization of many new and unusual *optical* properties”

Where removing the words regarding their specific focus (italicized) similarly describe our materials.

Reviewer comment

On balance, whilst the incorporation of negative stiffness within a 2-phase composite to achieve low stiffness (including quasi-static) is not new, making this a marginal assessment, the embodiment of this in nanowires and nanolaminates probably warrants publication of this manuscript in Nature Communications once the authors have addressed the issues raised above.

Authors response:

We thank the Reviewer for the analysis of our work and useful suggestions.

Reviewers' Comments:

Reviewer #1:

Remarks to the Author:

I am happy with the revised version of the manuscript and support its publication as is.

Reviewer #2:

Remarks to the Author:

The authors have addressed my comments to the best they can. Although it remains to be seen as if the proposed approach to reduce stiffness can be realized in reality, the manuscript can be published to stimulate further work on this.

Reviewer #3:

Remarks to the Author:

The reviewer thanks the authors for their careful consideration of the issues raised in the review of the original submission. The work is now set appropriately in terms of 'exceeding the bounds' and the effects of structure afforded by cellular materials, for example.

The authors have added reference to the paper by Hewage et al, as requested in the original review. However, the authors make an 'important distinction' relating to a lack of threshold stress (or lack of reduction in strength) in the authors' system. The authors need to be careful here. Hewage et al do not consider threshold stress in their model, although this is evident in the particular experimental systems (buckled beam and others) they report to validate the model. The current authors do not present any experimental validation for their model and so it is not known if there will be a threshold stress/reduction in strength associated with their concept in reality. It is probably advisable to remove this distinction in the current paper.

The other issue that remains unresolved is the use of the term 'metamaterial' to describe the system reported in the paper. In their rebuttal letter there are two definitions cited by the authors which make clear metamaterials derive properties based on structures composed of building blocks. On this we agree. However, the authors are referred to their own response relating to the Hewage system: "In contrast to our work, this [negative stiffness in the Hewage system] is achieved at the structural, not material, level ... ". The authors clearly consider their development a material not structural effect. Ergo it is not a metamaterial. If the authors persist in using this term they need to answer the following questions.

What is the defined structural 'building block', present in both definitions for a metamaterial cited by the authors in their rebuttal letter, in the authors' development - the atom?

What is it that makes their work different to the coherent nanostructures referred to in references 34-37 to justify the use of the metamaterial term? As distinct from, say, 'nanocomposite' (which seems a much more appropriate and correct term to apply)? References 34-37 consistently do not use the term 'metamaterial'.

Reply to Reviewer reports

Reviewer #1 (Remarks to the Author):

I am happy with the revised version of the manuscript and support its publication as is.

Reviewer #2 (Remarks to the Author):

The authors have addressed my comments to the best they can. Although it remains to be seen as if the proposed approach to reduce stiffness can be realized in reality, the manuscript can be published to stimulate further work on this.

Author's response:

We thank Reviewers 1 and 2 for their time and valuable input for improving the manuscript.

Reviewer #3 (Remarks to the Author):

Reviewer comment:

The reviewer thanks the authors for their careful consideration of the issues raised in the review of the original submission. The work is now set appropriately in terms of 'exceeding the bounds' and the effects of structure afforded by cellular materials, for example.

The authors have added reference to the paper by Hewage et al, as requested in the original review. However, the authors make an 'important distinction' relating to a lack of threshold stress (or lack of reduction in strength) in the authors' system. The authors need to be careful here. Hewage et al do not consider threshold stress in their model, although this is evident in the particular experimental systems (buckled beam and others) they report to validate the model. The current authors do not present any experimental validation for their model and so it is not known if there will be a threshold stress/reduction in strength associated with their concept in reality. It is probably advisable to remove this distinction in the current paper.

Author's response:

The reference to lack of threshold stress as an 'important distinction' in the manuscript refers to the comparison of our metamaterial nanowires and standard superelasticity, not with Hewage et al. We observe a threshold stress in the single component (martensitic) nanowire, indicating that threshold stress is, in fact, described by our model. With the same model, we do not observe a threshold stress in the lowest stiffness composites.

With respect to reduction in strength, nanolayered materials have been experimentally synthesized in many cases to specifically increase strength and hardness. We claim only to retain a similar strength in the metamaterials as compared to the separate components. The Reviewer is correct that no experimental data exists on the proposed metamaterials as they have not been fabricated yet; this is made clear in our manuscript.

Reviewer comment:

The other issue that remains unresolved is the use of the term ‘metamaterial’ to describe the system reported in the paper. In their rebuttal letter there are two definitions cited by the authors which make clear metamaterials derive properties based on structures composed of building blocks. On this we agree. However, the authors are referred to their own response relating to the Hewage system: "In contrast to our work, this [negative stiffness in the Hewage system] is achieved at the structural, not material, level ... ". The authors clearly consider their development a material not structural effect. Ergo it is not a metamaterial. If the authors persist in using this term they need to answer the following questions.

What is the defined structural 'building block', present in both definitions for a metamaterial cited by the authors in their rebuttal letter, in the authors' development - the atom?

What is it that makes their work different to the coherent nanostructures referred to in references 34-37 to justify the use of the metamaterial term? As distinct from, say, ‘nanocomposite’ (which seems a much more appropriate and correct term to apply)? References 34-37 consistently do not use the term ‘metamaterial’.

Author's response:

The definitions we cited in our previous response contain two criteria: structural building blocks and new, unusual properties beyond what is naturally possible. The building blocks in our materials are the individual layers (of differing composition), a nanoscale feature. Metamaterials often utilize micro and nanoscale features (at the material level), e.g. negative refraction optical metamaterials [1], in addition to macroscale approaches. Through our nanolayered structure we achieve unique, unexpected properties; therefore, our materials satisfy both criteria to be a metamaterial.

The cited nanostructures and related work are structurally similar to ours. Refs. 34-35 focus on techniques for creating the structures and devices, rather than unique properties. Ref 36 considers only the structural features of the nanolayers, while Ref. 37 shows the anisotropy possible through single layers. So none of these examples address the second criterion from the metamaterial definitions. Nevertheless, their work demonstrates the processes needed for creating material-level mechanical metamaterials experimentally.

Related work in nanolayered metals focuses on increasing strength and hardness. In these cases the expected bounds are not well defined and the effects depend much more on interfaces, rather than the combined response of the material overall, making it more difficult to consider those systems metamaterials.

1. Soukoulis, C. M. & Wegener, M. Past achievements and future challenges in the development of three-dimensional photonic metamaterials. *Nat. Photonics* **52**, 47–50 (2011).

We thank Reviewer 3 for thorough discussion and their additional time and analysis of our manuscript.